

# Rapid coral reef assessment using 3D modelling and acoustics: acoustic indices correlate to fish abundance, diversity and environmental indicators in West Papua, Indonesia

Mika Peck[1], Ricardo F. Tapilatu[2], Eveline Kurniati[3] and Christopher Rosado[3]

[1] School of Life Sciences, University of Sussex, Brighton, East Sussex, UK
[2] Marine Science Department and Research Centre of Pacific Marine Resources, University of Papua, Manokwari, West Papua, Indonesia
[3] Creative Action Tank, Jakarta, Indonesia

Corresponding author
Mika Peck, m.r.peck@sussex.ac.uk

## ABSTRACT

**Background:** Providing coral reef systems with the greatest chance of survival requires effective assessment and monitoring to guide management at a range of scales from community to government. The development of rapid monitoring approaches amenable to collection at community level, yet recognised by policymakers, remains a challenge. Technologies can increase the scope of data collection. Two promising visual and audio approaches are (i) 3D habitat models, generated through photogrammetry from video footage, providing assessment of coral cover structural metrics and (ii) audio, from which acoustic indices shown to correlate to vertebrate and invertebrate diversity, can be extracted.

**Methods:** We collected audio and video imagery using low cost underwater cameras (GoPro Hero7™) from 34 reef samples from West Papua (Indonesia). Using photogrammetry one camera was used to generate 3D models of 4 m² reef, the other was used to estimate fish abundance and collect audio to generate acoustic indices. We investigated relationships between acoustic metrics, fish abundance/diversity/functional groups, live coral cover and reef structural metrics.

**Results:** Generalized linear modelling identified significant but weak correlations between live coral cover and structural metrics extracted from 3D models and stronger relationships between live coral and fish abundance. Acoustic indices correlated to fish abundance, species richness and reef functional metrics associated with overfishing and algal control. Acoustic Evenness (1,200–11,000 Hz) and Root Mean Square RMS (100–1,200 Hz) were the best individual predictors overall suggesting traditional bioacoustic indices, providing information on sound energy and the variability in sound levels in specific frequency bands, can contribute to reef assessment.

**Conclusion:** Acoustics and 3D modelling contribute to low-cost, rapid reef assessment tools, amenable to community-level data collection, and generate information for coral reef management. Future work should explore whether 3D models of standardised transects and acoustic indices generated from low cost

underwater cameras can replicate or support 'gold standard' reef assessment methodologies recognised by policy makers in marine management.

## INTRODUCTION

Indonesia's coral reefs support exceptional biodiversity, providing food security and other important ecosystem services to many millions of people (*FAO, 2018*). Coral reefs face increasing anthropogenic threat, with loss of 23% of Indonesian corals between 1999 and 2011 (*Carter, 2018a*). Of remaining reefs, a third are considered 'good to excellent condition' with the remainder suffering various levels of anthropogenic degradation. The Birds Head Peninsula of West Papua and its reef ecosystems are recognised as the global epicentre of marine diversity, but a lack of resources and information currently limits evidence-based conservation action to address degradation of its reef ecosystems (*Carter, 2018b*). More optimistically, recent modelling identifies Indonesian reef systems of West Papua as belonging to 50 'bioclimatic reef units' having the highest probability of surviving bleaching impacts from predicted climate change (*Beyer et al., 2018*) that could cause loss of over 90% of global reefs (*Frieler et al., 2013*).

The key to providing Indonesia's reefs, and the populations dependent on them, the greatest chance of adaptation to climate change is by reducing proximate anthropogenic pressures (*Hughes et al., 2017*). These include overfishing, illegal and destructive harvest, habitat extraction, uncontrolled tourism and marine pollution. Effective management requires assessment, engagement and monitoring to determine impact of interventions, however the development of monitoring approaches amenable to collection at community level, yet recognised by policymakers, remains a challenge. Technologies can increase the scope of data collection in marine environments (*Obura et al., 2019*), but often require specialist training and equipment. Two promising visual and audio approaches amenable to rapid and community-level data collection for coral reef monitoring include (i) habitat metrics, such as rugosity, extracted from 3D models generated by photogrammetry (Structure from Motion SfM) (*Young et al., 2017*; *Burns et al., 2019*) and (ii) use of acoustic indices as a proxy for vertebrate and invertebrate diversity (*Bertucci et al., 2016*; *Bohnenstiehl et al., 2018*; *Bolgan et al., 2018*; *Obura et al., 2019*; *Elise et al., 2019b*; *Davies et al., 2020*).

Structural metrics, such as rugosity, provide a measure of physical complexity that underpins reef fish diversity and abundance (*Gratwicke & Speight, 2005*; *Alvarez-Filip et al., 2009*; *Raoult et al., 2016*; *Darling et al., 2017*). The principal method used by reef scientists to measure structural complexity is the chain-and-tape method, which produces a measure of rugosity calculated as the ratio of contour–following vs. straight distance between two points on the reef—resulting in the rugosity index ranging from 1 for a flat reef to rarely greater than 3 (*Alvarez-Filip et al., 2009*). Recently photogrammetry imagery

captured using underwater video of reef and rendered into a 3D models has allowed estimation of chain-and-tape rugosity and opened the opportunity for more complex measures of habitat structure, such as fractal dimension and vector dispersion, also shown to correlate to fish diversity (*Storlazzi et al., 2016*; *Young et al., 2017*; *Fukunaga et al., 2019*). These new methodologies, in addition to providing datasets for more complex studies of structure (*Burns et al., 2019*), provide the potential for archiving long-term repositories of 3D images of structure and coral diversity to provide habitat baselines for monitoring to guide management action or restoration activities (*Fukunaga et al., 2019*).

With habitat structure and coral cover/diversity information captured visually, acoustics can contribute by providing a proxy measurement for both diversity and function (*Elise et al., 2019b*). The coral reef environment has a unique soundscape (*Lobel, Kaatz & Rice, 2010*) generated by marine organisms relying on sound for a range of activities including navigation, spawning, feeding, mating, and avoiding predators (*Amorim, 2006*; *Tricas & Boyle, 2014*). Many reef fish species are known to produce sounds to attract mates, warn of danger, scare competitors and predators and maintain social cohesion (*Mann & Lobel, 1995*; *Tricas & Boyle, 2014*). The characteristic crackling sound of reefs is thought to reflect snapping shrimp (Family Alpheidae) that create broadband, high frequency, snaps in conspecific territorial interactions and feeding (*Versluis et al., 2000*). Acoustic indices are mutimetrics that allow the quick screening of complex acoustic data as prior knowledge of the composition of the acoustic community is not required— unlike automated analysis based on sound type detection and recognition requiring previous knowledge of targeted signals (*Vieira et al., 2015*), and time consuming manual analyses that require high levels of expertise. Some previous studies have shown promise, with correlations between acoustic indices and live coral cover (*Bertucci et al., 2016*; *Kaplan et al., 2018*; *Elise et al., 2019b*), fish abundance and diversity (*Kennedy et al., 2010*; *Bertucci et al., 2016*; *Staaterman et al., 2017*; *Kaplan et al., 2018*). Even relatively simple acoustic descriptors, such as the root-mean-square (RMS) of raw audio signal data have been shown to correlate well with percentage of living coral cover in tropical reefs suggesting the potential for development of low-cost acoustic habitat assessment tools for coral reef environments (*Bertucci et al., 2016*). Acoustic monitoring could provide a cost-effective means to remotely assess the community and even functional characteristics of specific marine habitats (*Elise et al., 2019b*) although care is needed at the different stages of their implementation to clearly understand what the metrics are responding to (*Bolgan et al., 2018*).

In this study we collected visual and acoustic data from the West Papuan Reef systems of Raja Ampat and Manokwari to address the following specific question; do habitat metrics and acoustic indices, extracted from underwater audio and video, correlate to coral cover, fish diversity and functional measures of ecological status?

## MATERIALS AND METHODS

Sampling took place from 10 reef sites in Raja Ampat (Waisal S 0° 26.417′, E 130° 44.418′, four replicates; Batu Lima S0° 27.010′ E 130° 41.807′, two replicates; Yenros S0° 27.624′S E 130° 41.451′ one replicate; Sawanare S0° 35.418′ E130° 36.209′, eight replicates;

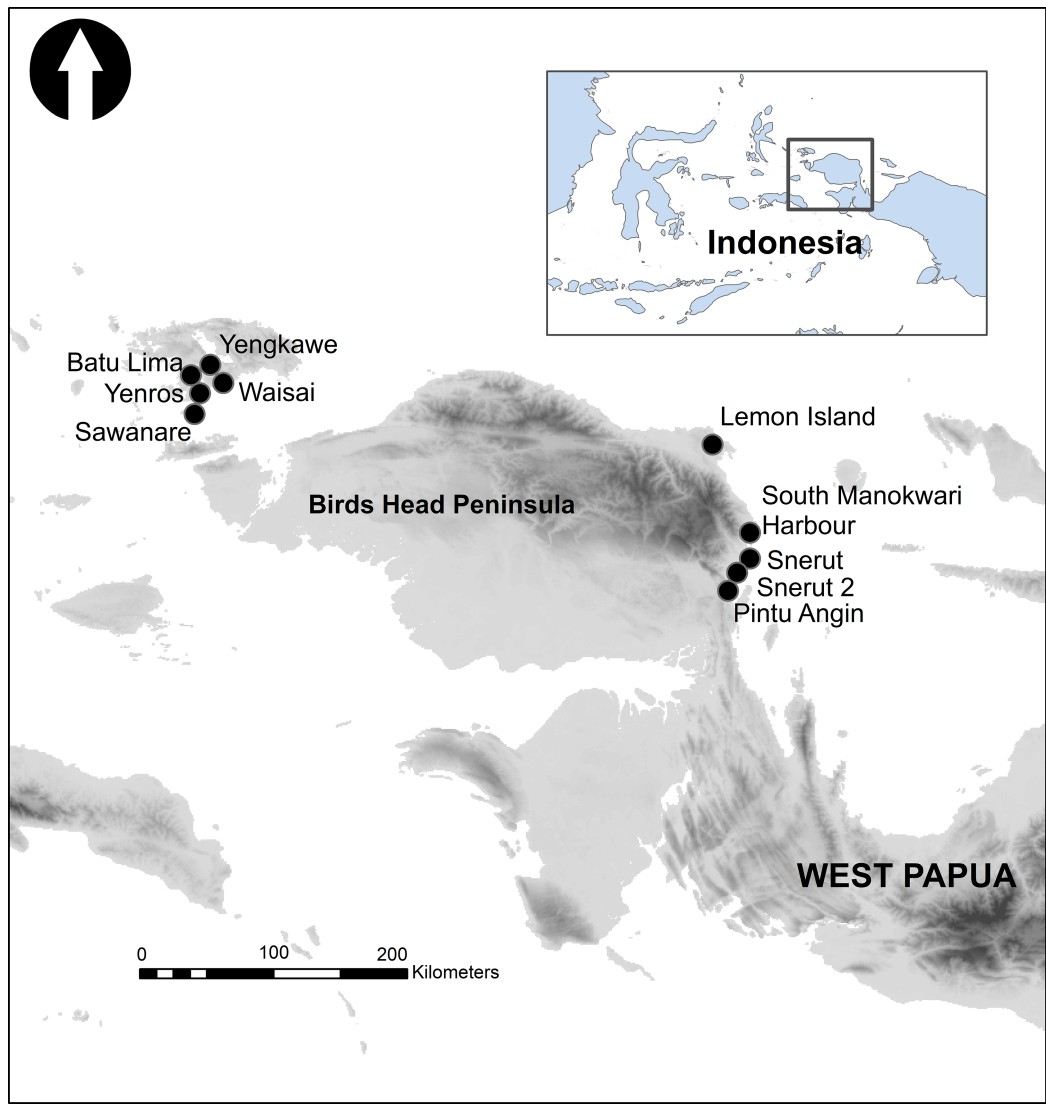

**Figure 1 The Birds Head Peninsula of West Papua Province, Indonesia, showing locations of sampling sites.**               

Yengkawe S0° 25.863′ E 130° 43.175′, five replicates) and Manokwari (Lemon Island S0° 53.317′ E 134° 4.714′, four replicates; Snerut S01 °28.211′ E134° 11.954′, three replicates; Snerut 2 S01° 22.602′ E134° 13.780′, two replicates; South Manokwari harbour S01° 20.892′ E 134° 16.078′; two replicates; Pintu Angin S01° 43.130′ E 134° 05.046′, three replicates) under Indonesian government RISTEKDIKTI permit 346/E5/E5.4/SIP/2019 (Fig. 1). All of 34 individual replicates were collected by the two same SCUBA divers (Peck/Rosado) representing reef from 3 to 8 m depth between the times of 9 am and 4 pm (*Elise et al., 2019a*). Following random choice of the initial quadrat at a site, subsequent replicates were generated at distances of 10 m at the same depth contour. For each 2 m × 2 m replicate, video was filmed in a 'lawnmower pattern' using a GoPro Hero7™ with a calibration object (Extended 2 m tape measure) included in imagery, as described by *Young et al. (2017)*. The camera was set to 1,920 × 1,080 resolution,
30 frames per second and maximum zoom to minimise fisheye effect. Once replicate imagery was collected a second GoPro Hero7™ camera, mounted to a weight and set to default fish-eye, was placed in a corner of the quadrat pointed seaward to capture video of fish communities and record audio over a 10-min period as both SCUBA divers moved at least 10m away to avoid disturbance to recordings and fish.

Generation of 3D models followed protocol in *Young et al. (2017)*. In summary, raw video footage was first converted to overlapping image sequences every 10 frames (3fps) using free software FFmpeg (www.ffmpeg.org) for import to Agisoft Metashape (https://www.agisoft.com/). The model was rendered into 3D imagery then exported as a wavefront (.obj) file for analysis using Rhinoceros (https://www.rhino3d.com/) and extraction of rugosity metrics, rugosity, vector dispersion and fractal dimension (*Young et al., 2017*). Overhead images of the reef transect were exported from Agisoft Metashape as jpeg files for estimation of percentage live coral cover in ImageJ (https://imagej.nih.gov/ij/). Total abundance of fish, abundance of fish estimated at >30 cm and fish species richness was estimated by an experienced ichthyologist (author EK) from video from 5 to 6 min (this was chosen as it allows time for fish to settle after camera placement and divers to retreat to avoid acoustic interference). Coral health was assessed by the number of Chaetodontidae (Butterflyfish), fishing pressure estimated by summing counts of Serranidae (Groupers), Lutjanidae (Snapper) Lethrinidae (Emperors) and Haemulidae (Grunts), with a measure of Algal control provided by a count of Scaridae (Parrotfish), Acathuridae (Surgeonfishes, tangs, unicornfish) and Siganidae (Rabbitfish) (*Giyanto, Dhewani & Abrar, 2017*). Motor noise from each minute of video sampled was also reported as present or absent.

Audio was extracted from videos using Audacity software (https://www.audacityteam.org/) and saved as WAV files (Stereo, project rate 48,000 Hz, 32-bit float) for further analysis. Two approaches to acoustic analysis were undertaken. The first applies generalized linear modelling (GLM) to investigate relationships between fish diversity metrics, live coral cover, structural complexity (rugosity and vector dispersion) and acoustic indices. The Soundecology package (multiple_sounds; *Villanueva-Rivera & Pijanowski, 2018*) was used to extract acoustic indices from audio (min 4–5, 5–6, 6–7) for two frequency ranges. A lower band representing 'fish' audio of 100–1,200 Hz (100 Hz bins) unless otherwise stated, and a higher frequency range, 1,200–11,000 Hz (1,000 Hz bins), representing 'invertebrates' (*Patrick Lyon et al., 2019*). Acoustic Complexity Index (ACI; FFT window = 512, cluster size = 5 s, left hand channel) (*Pieretti, Farina & Morri, 2011*) Acoustic Diversity Index (ADI; bin frequency step = 100 Hz, 0–1,200 Hz/ 1,200–11,000 Hz, left channel) (*Villanueva-Rivera et al., 2011*), Acoustic Evenness Index (AEI db threshold −50 db, frequency step = 100 for 0–1,200 Hz, frequency step = 1,000 Hz for 1,200–11,050 Hz) (*Villanueva-Rivera et al., 2011*) and Bioacoustic Index (BI; FFT window = 512, left channel) (*Boelman et al., 2007*) were calculated for the left channel and both frequency ranges). RMS and Roughness were calculated for each frequency band using R package seewave (*Sueur, Aubin & Simonis, 2008*). Ratios (Acoustic index at low frequency range 100–1,200 Hz/acoustic index at frequency range 1,200–11,050 Hz) were calculated for amplitude-based indices (RMS, Roughness, ACI, BI) to avoid influence of auto gain in GoPro acoustic recordings extracted from video (*Lindseth & Lobel, 2018*).

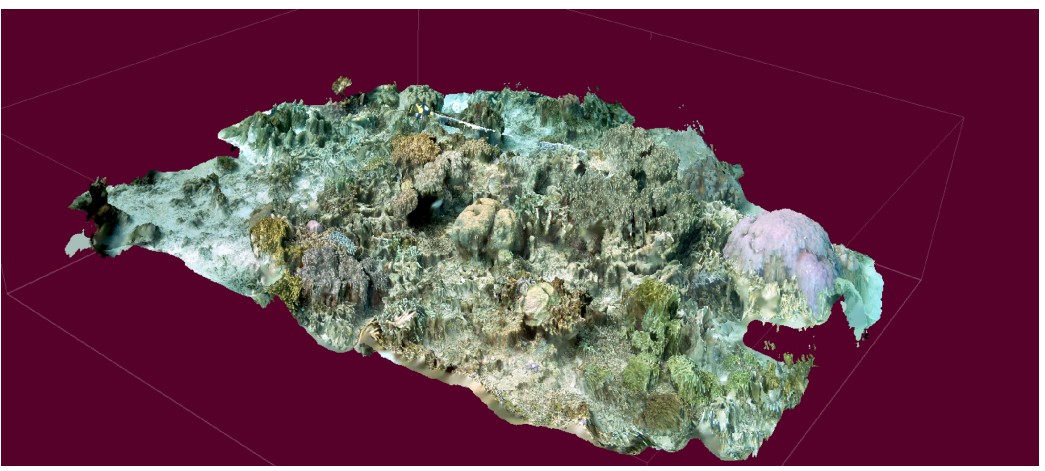

**Figure 2  3D model reconstructed from video imagery.**

Values for both low and high frequency ranges and ratios were compared for spectral frequency-based indices (ADI, AEI). Temporal stability of acoustic indices was explored using correlation analysis (R package vegan).

Generalised linear models were used to explore relationships between; fish abundance, species richness and fish community indicators against live coral cover (Poisson model), live coral cover and acoustic indices (Binomial model), fish diversity indices and acoustic indices (Poisson models), rugosity and acoustic indices (Gaussian) and vector dispersion and acoustic indices (Gaussian). Models were visually checked for normality and homoscedasticity of residuals. Model selection was applied to identify meaningful relationships based on Akaike's Information Criterion (AIC), with 'percentage deviance explained' (1-deviance/null model deviance) reported.

The second approach investigated covariate relationships with mean frequency spectrum for each audio sample. Distance based redundancy analysis (db-RDA) was applied to the cumulative spectral dissimilarity matrix of the mean frequency spectra (*Lellouch et al., 2014*) (FFT Hanning window = 512, left channel) for both the lower frequency range (100–1,200 Hz) and higher range (1,200– 11,050 Hz), following *Sueur (2018)* , with Monte Carlo testing for significance of covariates. All statistical programming was undertaken in RStudio Version 1.2.5033 (*RStudio Team, 2015*) using R (Version 3.6.2) and code and datasets are available in GitHub at https://github.com/mrp21/West-Papua. The repository also contains links to sample audio and video.

## RESULTS

Extraction of acoustic indices was successful for all replicates, with rendering of 3D imagery and extraction of subsequent rugosity measures successful in 31/34 replicates (Fig. 2).

A significant positive relationship ($Z_{30, 29} = 50.82$, $p < 0.0001$), showing an increase in fish abundance with an increase in live coral cover, is observed for the GLM Poisson model of fish abundance count data against proportion of live coral cover, accounting for 47.2% of deviance (Fig. 3). All other diversity metrics for fish are non-significant ($p > 0.05$)

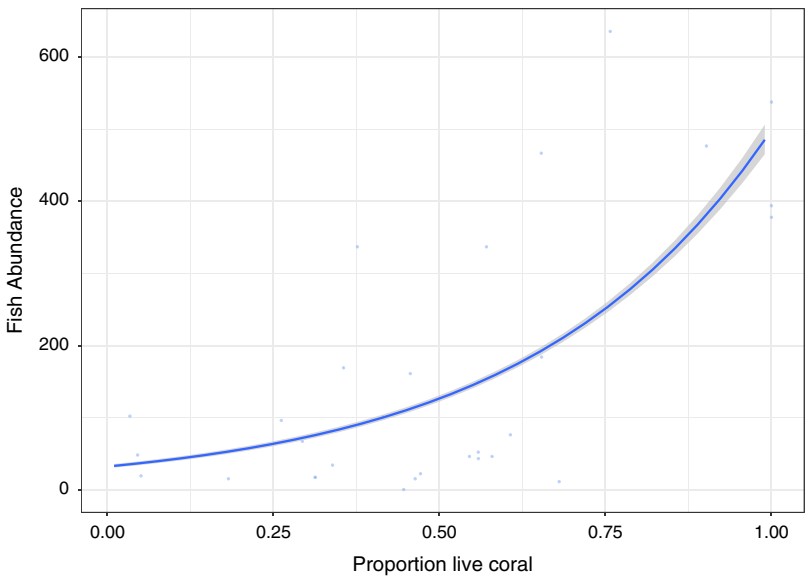

**Figure 3 Fish abundance against percentage live coral cover.**

in relation to live coral cover. Rugosity ($Z_{30, 29} = 13.54$, $p < 0.0001$) and Vector dispersion ($Z_{29, 28} = -4.27$, $p < 0.0001$) are statistically significant but only explain 3.4% and 0.3% of variability in live coral cover respectively.

Temporal stability across replicates for acoustic indices extracted from min 4 to 5, 5 to 6 and 6 to 7 is shown in Fig. 4 (Supplemental Information S1), with eight of the acoustic indices showing temporal correlation coefficients of over 0.7. The stability of metrics ranged from a high correlation coefficient of 0.95 for ACI Ratio to 0.37 AD Ratio. To guide interpretation, the correlation coefficient for fish abundance across this same timescale for all replicates was 0.74.

A GLM model including all acoustic indices was not significantly different from a null model in explaining proportional coral cover (LHR $_{30, 20} = 2.74$, $p = 0.98$). Acoustic indices did not significantly explain rugosity ($F_{30, 20} = 0.08$, $p = 0.61$) or vector dispersion ($F_{29, 19} = 0.02$, $p = 0.89$), suggesting no direct link between acoustics and structure at the scale of this study.

Using all 18 acoustic indices (extracted from min 5 to 6; sample spectrograms in Fig. 5) to model fish abundance explains 83.4% of model deviance compared to a null model ($\chi^2_{33, 15} = 5,999.1$, $p = < 0.0001$). A significant difference between this model and one that includes a marker for motor noise ($\chi^2_{14, 15} = -91.194$, $p = < 0.0001$) meant that we removed seven samples with motor noise to better understand the ability of acoustic indices to respond to marine acoustic environments without obvious anthrophony in all analyses below. Without obvious motor noise the 18 indices significantly explained 89.4% of model deviance ($\chi^2_{26, 8} = 2,875.8$, $p = < 0.0001$). Forward, backwards and bothway covariate selection, using Akaike's Information Criteria (AIC) to select the most parsimonious model, suggests fish abundance can be modeled as effectively (89.4% deviance explained) as the full model using 16 predictors although differences in AIC between models is small.

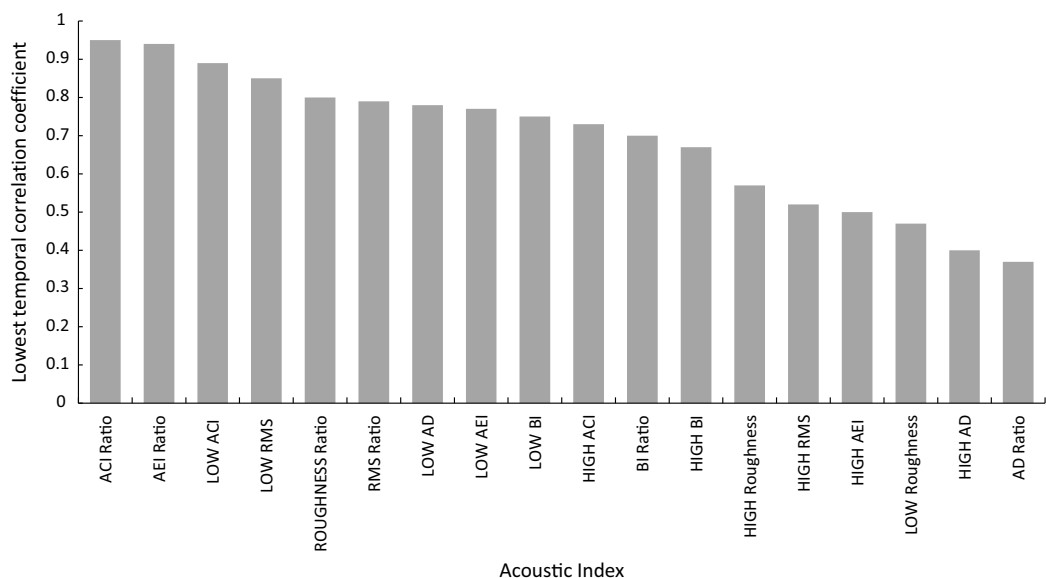

**Figure 4 Shorth term temporal stablility of acoustic indices illustrated by correlation coefficient of index across min 4–5, 5–6 and 6–7.**

Significant parsimonious models based on lowest AIC values show that fish species richness is significantly explained by four acoustic indices ($\chi^2_{26,\,8}$ = 342.1, $p$ =< 0.0001; 38.6% deviance explained), Fish >30 cm length is predicted by 14 acoustic indices ($\chi^2_{26,\,8}$ = 341.5, $p$ =< 0.0001; 87.3% deviance explained), and Fishing pressure is predicted by four indices ($\chi^2_{26,\,8}$ = 19.7, $p$ = 0.0005; 34.5% deviance explained). Algal control is best predicted by 16 indices ($\chi^2_{26,\,8}$ = 116.2, $p$ =< 0.0001; 81.3% deviance explained) and Coral Health by 4 indices ($\chi^2_{26,\,8}$ = 12.9, $p$ = 0.01; 27.3% deviance explained).

Generalized linear modelling analysis identifies all individual acoustic indices as statistically significant in explaining fish abundance, with deviance explained ranging from 0.2% to 40% (Table 1; Supplemental Information S2). The top three indices explaining fish abundance are AEI (1,200–11,050 Hz; 40% deviance explained), ACI (100–1,200 Hz; 38% deviance explained) and RMS Ratio (33.5% deviance explained). The top three indices predicting fish species richness are AEI (1,200–11,050 Hz; 24% deviance explained), ADI (1,200–11,050 Hz 19% deviance explained) and AEI Ratio (19.2% deviance explained). For fish over 30 cm in length the top predictors are ACI (1,200–11,050 Hz; 25.3% deviance explained), RMS (100–1,200 Hz; 18.3% deviance explained) and AEI Ratio (17.8% deviance explained). Fishing Pressure is predicted by a similar set of indices; RMS (100–1,200 Hz; 12% deviance explained), ACI (1,200–11,050 Hz; 10.9% deviance explained) and AEI (1,200–11,050 Hz; 7.3% deviance explained). Algal control is predicted best by AEI Ratio (22.9% deviance explained), AEI (1,200–11,050 Hz; 21.8% deviance explained) and RMS (100–1,200 Hz; 21.7% deviance explained). Coral Health (Corallivores) was not significantly predicted by any index.

No single acoustic indicator provides the best prediction for all categories however AEI (1,200–11,050 Hz) provides top three predictions for four dependent variable categories (Abundance, Species richness, Fishing Pressure and Algal Control) and the highest for
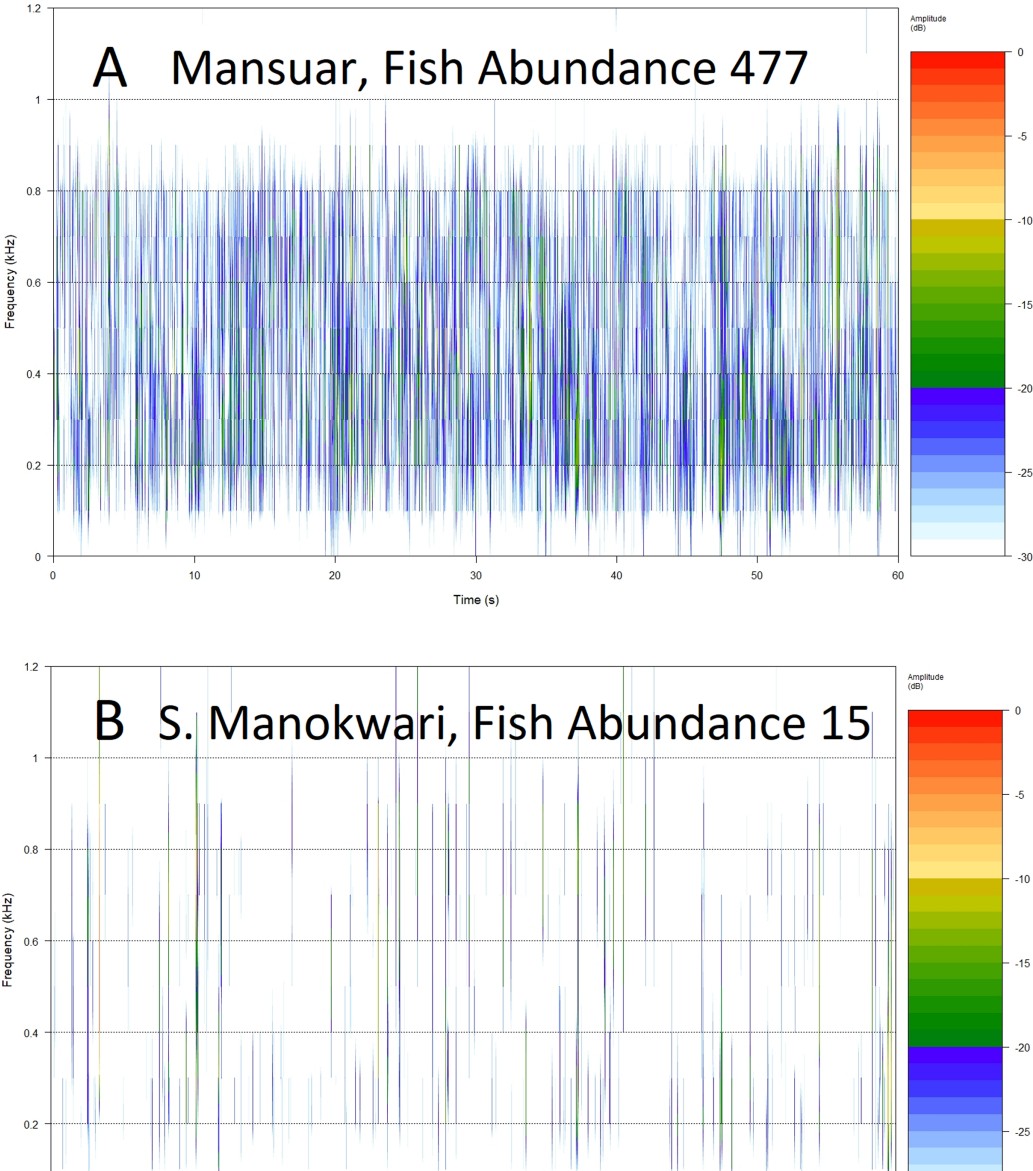

**Figure 5 Example spectrograms of lower frequency band (100–1,200 Hz) from replicates representing high (A) and low (B) fish abundance.**

fish abundance (Fig. 6). RMS (100–1,200 Hz) provides top three predictions for three dependent variable categories (Fish over 30 cm, Fishing pressure and Algal control), and the highest for indicator of fishing pressure although it has low explanatory power. All other acoustic indices provide only top three predictions for two or fewer dependent variable categories (Table 1).

A correlation plot, for model deviance explained by all acoustic indices across fish abundance, species richness and reef status indicator categories (Fig. 7), shows that

**Table 1 Deviance explained by individual acoustic indices for fish diversity and environmental indicator groups.** Significant deviance explained by GLMs is highlighted in bold and grey highlights identify top 3 acoustic indicator predictors for fish diversity measures and environmental indicators.

| GLM model acoustic index predictor | Deviance explained for all statistically significant models | | | | | |
|---|---|---|---|---|---|---|
| | Fish abundance (%) | Fish >30 cm length (%) | Coral health (Corallivores) (%) | Fishing pressure (targeted fish families) (%) | Algal control (fish algal control) (%) | Fish species diversity (%) |
| AEI HIGH | 40.0 | 11.3 | 0.0 | 7.3 | 21.8 | 24.1 |
| AC LOW | 38.4 | 17.5 | 0.0 | 0.0 | 14.0 | 10.4 |
| RMS Ratio | 33.5 | 13.2 | 0.0 | 0.0 | 20.3 | 14.0 |
| AC HIGH | 33.0 | 25.3 | 0.0 | 10.9 | 8.3 | 11.3 |
| AC Ratio | 31.7 | 8.8 | 0.0 | 0.0 | 14.7 | 6.6 |
| AD HIGH | 31.1 | 6.7 | 0.0 | 0.0 | 17.3 | 19.4 |
| BI HIGH | 23.6 | 13.8 | 0.0 | 0.0 | 7.7 | 6.3 |
| AEI Ratio | 23.3 | 17.8 | 0.0 | 6.8 | 22.9 | 19.2 |
| AD Ratio | 22.8 | 3.7 | 0.0 | 0.0 | 13.2 | 15.3 |
| HIGH RMS | 21.2 | 2.1 | 0.0 | 0.0 | 3.8 | 5.0 |
| BI Ratio | 18.7 | 16.2 | 0.0 | 0.0 | 16.5 | 4.4 |
| LOW RMS | 17.7 | 18.3 | 0.0 | 12.0 | 21.7 | 6.9 |
| BI LOW | 8.5 | 11.4 | 0.0 | 0.0 | 15.3 | 2.7 |
| HIGH ROUGHNESS | 6.9 | 0.0 | 0.0 | 0.0 | 9.7 | 8.0 |
| ROUGHNESS Ratio | 2.1 | 0.0 | 0.0 | 0.0 | 7.6 | 2.5 |
| AD LOW | 1.6 | 7.8 | 0.0 | 0.0 | 4.4 | 2.1 |
| LOW ROUGHNESS | 1.3 | 2.5 | 0.0 | 0.0 | 6.1 | 0.4 |
| AEI LOW | 0.2 | 4.6 | 0.0 | 0.0 | 3.9 | 0.8 |

good explanatory power for acoustic indices for larger fish (over 30 cm in length) is positively correlated to indicator species of fishing pressure (Correlation coefficient 0.64; $p = 0.004$) and fish abundance (Correlation coefficient 0.56; $p = 0.01$,). Explanatory power for fish abundance is also significantly correlated to explanatory power for indicator species of coral reef algal control (Correlation coefficient 0.57; $p = 0.01$). Fish species richness, explained by acoustic indices, is correlated with abundance (Correlation coefficient 0.75; $p < 0.001$) and indicator species of coral reef algal control (Correlation coefficient 0.68; $p = 0.002$).

Distance based redundancy analysis show the first axis explains 63% of projected inertia based on spectral dissimilarity for frequencies 100–1,200 Hz with only site showing significant explanatory power (Monte-Carlo, 1000, $p < 0.001$), explaining 72.2% of inertia. All other locational, structural or fish species richness covariates provide no explanatory power ($p > 0.05$). For the higher frequency range (1,200–11,050 Hz) a similar pattern is seen with the first axis providing 68% of projected inertia with only site showing significant predictive power (Monte-Carlo, 1000, $p = 0.02$) and explaining 54.4% of inertia. It is worth noting that fish abundance is nearly significant (Monte-Carlo, 1000, $p = 0.06$), explaining 93.3% of inertia although all other predictors are statistically insignificant.

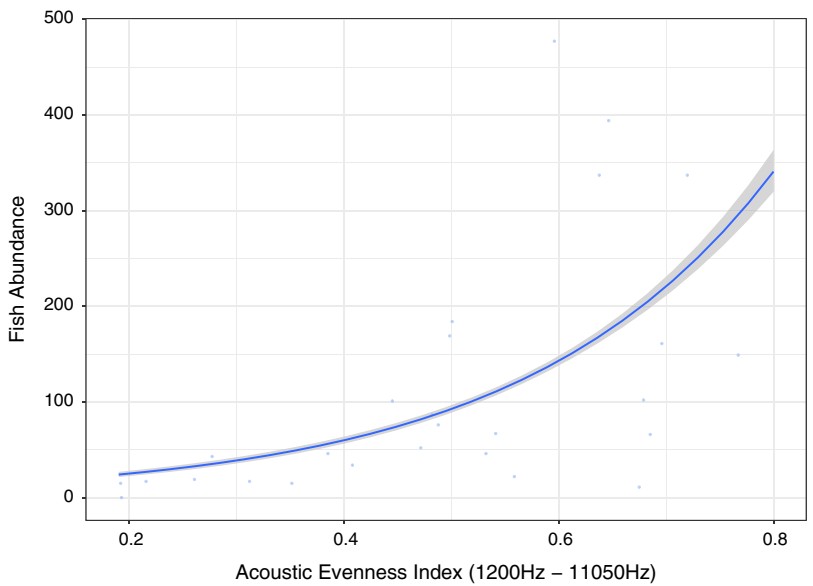

**Figure 6 Fish abundance against acoustic evenness index (1,200–11,050 Hz).**

## DISCUSSION

Effective management of extensive and remote coral reef systems, such as those of the Birds Head Seascape in West Papua, requires engagement with people from local communities in assessment, management and monitoring activities. However, coral reef assessment and monitoring methodologies amenable to data collection at grassroots level are rarely recognised by policymakers, as they can lack methodological and statistical rigour of standard methods (*Obura et al., 2019*). Development of tools to collect reef status data applicable to citizen, or civic science projects (*Dillon, Stevenson & Wals, 2016*; *Schmiedel et al., 2016*), yet recognised by policymakers could increase the scope of effective reef management. Technologies can provide the opportunity to increase scope and rigour of data collected, and it is within this context that we investigated the use of relatively low-cost underwater cameras in assessment of coral reef status. We used simple video protocols to collect fish abundance estimates, acoustic records and structural habitat metrics through subsequent photogrammetric 3D reconstruction of reef quadrats (*Bertucci et al., 2016*; *Patrick Lyon et al., 2019*). Each individual 10-min replicate could be collected by two people in approximately 15 min, an important consideration if using SCUBA equipment that limits dive times. Estimation of abundance and species richness of fish from video footage does, however, require time and expertise in fish identification and good consistent placement of cameras to ensure similar field of views are available for analysis. The extraction of 3D models also requires technical skill, access to computers and time, as does the extraction of audio and analysis of acoustic indices, although future development of apps could minimise the technical burden on users. To our knowledge this is the first time audio from low-cost cameras has been investigated to support short-term reef assessments, with previous studies requiring simultaneous deployment of separate
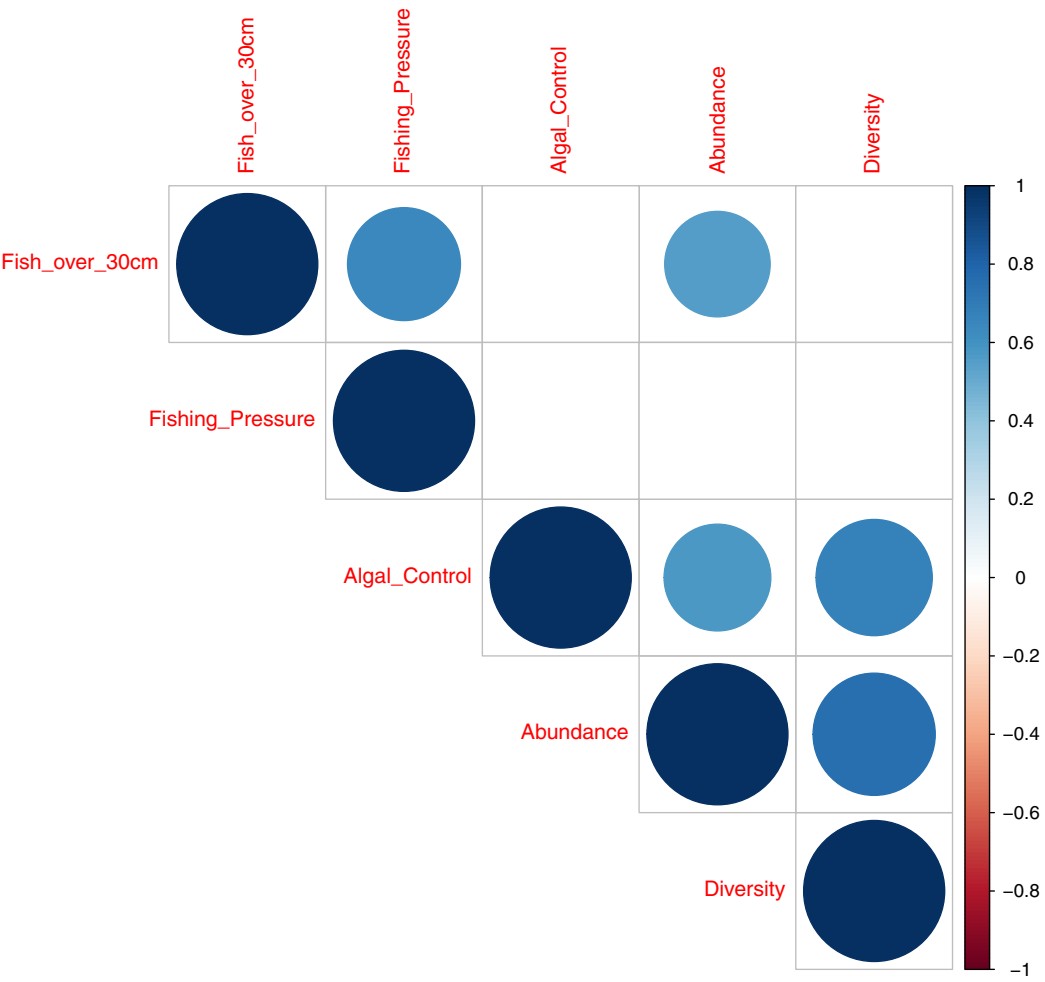

**Figure 7 Correlation plot illustrating deviance explained by acoustic indices across fish diversity and environmental indicator groups.**

hydrophone equipment, although it should be noted that the GoPro™ microphone is not as sensitive or omnidirectional as a dedicated hydrophone, and might underestimate vocal community complexity.

Fish abundance correlated significantly with proportion of hard coral cover reflecting similar correlations between abundance and coral cover in studies carried out at a similar scale (*Komyakova, Munday & Jones, 2013*). This relationship highlights the importance of live coral, and its measurement, in reef assessments, although coral structural forms may be more important predictors of functional or biodiversity status (*Elise et al., 2019b*). With coral playing roles as a food source for corallivores, as shelter from predation and for reproduction for reef fish it is clear that differing coral morphologies support a range of different functional requirements dependent on fish species and life strategies (*Komyakova, Munday & Jones, 2013*).

Structural complexity is recognised as an integral component of coral reef ecosystems correlating to total live coral cover, fish density and biomass (*Graham & Nash, 2013*), however the strength and evidence for correlations have varied between studies

(*Alvarez-Filip et al., 2011*; *Graham & Nash, 2013*; *Komyakova, Munday & Jones, 2013*; *Burns et al., 2019*). We found a significant but weak correlation between fish abundance and rugosity and vector dispersion metrics extracted from reconstructed 3D imagery of reef (*Young et al., 2017*), although this may reflect the smaller scale (4 m$^2$) of reef habitat analysed (*Fukunaga et al., 2019*). *Hernández-Landa, Barrera-Falcon & Rioja-Nieto (2020)* suggest 374 m$^2$ as the minimum sampling area to capture 90% of coral species richness for Caribbean systems with scale likely to play a major role in any analyses of this type. Previous studies have shown a lack of correlation between coral cover and structural complexity when coral cover is lower (*Graham et al., 2009*) and this may have weakened relationships in our study. Although total coral cover is often associated with greater structural complexity, *Burns et al. (2019)* show that different morphological types, that is branching *Acropora* and *Porites* corals, influence complexity in different ways suggesting future work should aim to better understand relationships between structural complexity and function at the level of coral genera/typology. It is, however, clear that reef degradation from tropical storms, bleaching events and declines in herbivores are associated with structural simplification (*Alvarez-Filip et al., 2009*), and influence structure of fish assemblages, making measurement of structural complexity a vital component of reef assessment protocols (*Fukunaga et al., 2019*). Photogrammetric techniques using low cost cameras do provide the opportunity for long-term archiving of 3D reef imagery and monitoring of structural degradation or recovery (*Graham & Nash, 2013*; *Obura et al., 2019*) whilst providing datasets for further investigation.

Differences in mean spectra for both the low frequency range and high frequency range between sites using the ß index, spectral dissimilarity (*Sueur, 2018*), could not be explained by any of the explanatory variables. Mean spectral values are calculated in the frequency domain and scaled to cumulative distribution functions in the comparison, so should be comparable across sites, even with variable gain measurements applied across samples. Our lack of correlation likely reflects the complexities of comparing frequency distributions even when dividing spectra into more meaningful ranges, as in our study, and could still show promise with further refinement.

The influence of habitat structure on soundscape has been seen in other work (*Freeman et al., 2018*; *Elise et al., 2019b*), suggesting that structurally complex environments accommodate higher diversity of soniferous fish species generating low frequency sounds. In our study we did not find a significant correlation between acoustic indices and structural measures, rugosity and vector dispersal, extracted from photogrammetry. This may reflect the scale of our study that focused on smaller 4 m$^2$ reef images that may not represent the broader soundscape or habitat complexity. *Elise et al. (2019b)* also suggest care needs to be taken to disentangle low frequency acoustic signals generated locally from sounds reflected by hard reef structures generated at a distance that propagate long distances underwater (*Lugli, 2012*). Although several other studies have found a correlation between acoustic indices and live coral cover (*Bertucci et al., 2016*; *Kaplan et al., 2018*; *Elise et al., 2019b*) we did not find a significant relationship. The relationship between live coral and reef acoustics may be more complex and depend more on coral growth form.
It is growth form that generates the habitat structures used by the range of soniferous vertebrate and invertebrate communities (*Elise et al., 2019b*; *Fukunaga et al., 2019*).

As with other studies (*Kennedy et al., 2010*; *Bertucci et al., 2016*; *Staaterman et al., 2017*; *Kaplan et al., 2018*), we observed some of the strongest relationships between acoustic indices and fish abundance. Indices providing explanatory power in our study can be grouped into those that measure aspects of complexity across frequencies (acoustic evenness), time (acoustic complexity) and proxies of energy levels (root mean square).

Measuring differences in frequencies, the index providing best explanatory power of fish abundance and relatively high explanation for fish species richness, fishing pressure and algal control was Acoustic Evenness (1,200–11,000 Hz) (Fig. 6; Table 1). Acoustic Evenness Indices apply the Gini coefficient (a measure of distribution inequality) to frequency bins over the sampling time period (*Villanueva-Rivera et al., 2011*). Working in the frequency domain this index should be robust to influence of auto gain functions associated with audio, allowing direct comparison of AEI index values across replicates investigated in this study. According to the literature the higher range AEI values (1,200–11,050 Hz) are associated with invertebrate activity (*Patrick Lyon et al., 2019*) with dominant invertebrate acoustic sound on coral reef environments thought to be generated through cavitation by snapping shrimp (*Versluis et al., 2000*). This distinctive sound, that peaks between 4 kHz and 6 kHz (*Au & Banks, 1998*), is put to use by Indonesian fishers to identify good fishing grounds above coral reef by placing their ear to a wooden oar lowered into the sea to listen for the 'crackling' sound (Y. Yahya, 2020, personal communications). Lower AEI values represent more even distribution of sound across frequency bins, with values approaching unity representing extreme contributions from one or a few frequency bins (*Villanueva-Rivera & Pijanowski, 2018*; *Lindseth & Lobel, 2018*). Low Acoustic Evenness in the higher frequency band at low fish abundance suggest more even distribution of sound across frequencies. With increase in fish abundance, greater variability in distribution is observed reflecting sound associated with greater fish vocalizations and/or reflects increasing invertebrate activity, with the index reporting increased 'crackling' associated with reef. This index may be reflecting increasing invertebrate activity that provides a proxy for more pristine ecosystem status with subsequently higher fish abundance, species richness and size.

Root Mean Square (RMS 100–1,200 Hz) provides a measure of sound energy within the system and is the best predictor of fishing pressure, that is the presence of target species within families Serranidae, Lutjanidae, Lethrinidae and Haemulidae (*Giyanto, Dhewani & Abrar, 2017*). It also provides relatively good predictive power for fish over 30 cm in length, and species associated with algal control. The relationship is visualized in Fig. 5, with more spectral bands and higher intensities contributing to the acoustic environment at sites with higher fish abundance (Figs. 5A and 5B). Soniferous fish are generally thought to contribute to the soundscape in this lower frequency range (*Kennedy et al., 2010*; *Elise et al., 2019b*) and this index could reflect increasing vocalizations from acoustically active fish species at these sites. These results agree with those observed by *Kaplan et al. (2018)* for Maui reef systems who found correlation between diel SPL (RMS) trends and soniferous fish. Some care should be taken in interpreting higher activity in the lower

frequency range however, as although soniferous fish contribute to the soundscape in the lower frequency range (*Kennedy et al., 2010*; *Elise et al., 2019b*) this frequency range also registers anthropogenic activity that has been seen to mask biophony (*Bolgan et al., 2018*). In this study we avoided obvious anthrophony by excluding files with sounds of motors, but it would be useful to develop a measure of anthrophony itself, as it can directly impact on a range of reef biological processes (*Simpson et al., 2005*; *Ruiz et al., 2017*; *Jain-Schlaepfer et al., 2018*), and act as a proxy for other anthropogenic pressures.

Our results suggesting AEI and RMS as the best predictors of fish abundance reflect the conclusions of *Kaplan et al. (2018)* that traditional bioacoustic indices providing information on sound energy and the variability in sound levels in specific frequency bands are easier to understand and more robust than other indices such as acoustic complexity index (ACI). In summary, acoustic indices providing information on both sound complexity (AEI) and energy (RMS) significantly correlate with fish abundance, species richness and family level indicators of ecosystem status potentially contributing to rapid bioassessment toolkits as suggested for terrestrial environments (*Eldridge et al., 2018*), although further testing and validation is required. It would be important to consider temporal variability of acoustic indices, as they differed significantly between indices investigated and further experimentation and analysis is required to determine what this implies in terms of choosing an optimal coral reef bioassessment acoustic index. Ultimately it is important to clearly define the direct or indirect acoustic linkage between acoustic index response and the ecological metric under investigation and undertake further work to tune and optimize acoustic indices (*Bolgan et al., 2018*). It should also be noted that diel and longer-term trends in acoustics cannot be captured using the approach outlined here and that future work on rapid acoustic assessment does need to place the measurement of rapid assessment acoustic indices in context of site-specific, longer-term, temporal patterns.

This work provides evidence that acoustic indices can generate information on the ecological status of reef environments. This is best illustrated in Fig. 7 that summarises the explanatory power of acosustic indices across covariates of fish abundance, species richness, and environmental indicator groups. Good explanatory power of fish abundance correlates well with explanatory power of fish species richness, fish over 30 cm in length, and species associated with algal control - counts of Scaridae, Acathuridae and Siganidae (*Giyanto, Dhewani & Abrar, 2017*). Good explanatory power for fish over 30 cm correlates with species targeted by fishers; Serranidae, Lutjanidae, Lethrinidae and Haemulidae (*Giyanto, Dhewani & Abrar, 2017*) and explanatory power for fish species richness correlates significantly with species associated with algal control. The results suggest a role for the use of acoustic indices as a measure of fish abundance and reef status as they correlate with multiple acoustic indices, with the best single acoustic index observed to be the Acoustic Evenness Index for the frequency range 1,200–11,000 Hz.

## CONCLUSIONS

The data collection process is amenable to community-level application with some training, and would allow use of snorkeling equipment rather than SCUBA, for shallow

reef systems. Each sample can be collected quickly (approx. 15 min per sample) with imagery providing information on coral cover and coral taxonomic diversity. 3D visual imagery provides the opportunity for long-term archiving of permanent transects and assessment of standard measures of coral cover and diversity and could play an important role in monitoring degradation, and recovery, of reef structures, both natural and artificial (*Graham & Nash, 2013*; *Darling et al., 2017*; *Fukunaga et al., 2019*; *Obura et al., 2019*). Acoustic indices offer insights to the status of biodiversity and function in reef environments, yet they remain rather blunt instruments. Greater insights might be gained through more advanced analytical approaches such as; sinusoidal modelling encompassing spectro-temporal space (*Eldridge et al., 2016*), identification of soniferous 'indicator' species of reef status, the development of acoustic taxonomies (*Desiderà et al., 2019*) and/ or linking acoustics more directly to components of reef function (*Elise et al., 2019b*). Future work should also explore whether 3D models of standardised transects and acoustic indices generated from low cost underwater cameras can replicate or support 'gold standard' reef assessment methodologies. This would bridge the current gap between data collected by community-based protocols and standard techniques that feed in to monitoring, management, governance and policymaking.

## ACKNOWLEDGEMENTS

We would like to thank the Bupati of South Manokwari, Marcus Waran and his staff for supporting field surveys in South Manokwari, staff of the University of West Papua, in particular Dimas Algutomo, for supporting the dive operations and Scuba Republic for logistics during reef survey of Raja Ampat. We thank Mohammad Abrar (LIPI, Indonesian Institute of Sciences) for project support and RISTEK for support in attaining the research and training visa (Visa 2C11DF0016AT).

### Funding

This work was supported by Research England (GCRF) International Challenge Development Fund (University of Sussex) Projects IDCF1-011 (Community-based data driven decision making for sustainable management of marine resources in Indonesia) & IDCF2-005 (Integrating anthropological methods and innovative research tools for community-based coral reef and fisheries conservation in West Papua, Indonesia). The funders had no role in study design, data collection and analysis, decision to publish, or preparation of the manuscript.

### Grant Disclosures

The following grant information was disclosed by the authors:
Research England (GCRF) International Challenge Development Fund (University of Sussex) Projects: IDCF1-011 and IDCF2-005.

## Competing Interests

Christopher Rosado is employed by Creative Action Tank. The authors declare that they have no competing interests.

## Author Contributions

- Mika Peck conceived and designed the experiments, performed the experiments, analyzed the data, prepared figures and/or tables, authored or reviewed drafts of the paper, and approved the final draft.
- Ricardo F. Tapilatu conceived and designed the experiments, performed the experiments, authored or reviewed drafts of the paper, administrative support, research visa, and approved the final draft.
- Eveline Kurniati analyzed the data, authored or reviewed drafts of the paper, and approved the final draft.
- Christopher Rosado conceived and designed the experiments, performed the experiments, authored or reviewed drafts of the paper, and approved the final draft.

## Field Study Permissions

The following information was supplied relating to field study approvals (i.e., approving body and any reference numbers):

Field research was approved by Indonesian government RISTEKDIKTI (346/E5/E5.4/SIP/2019).

## Data Availability

Raw data, including acoustic indices and habitat structural measurements extracted from 3D imagery, is available in the Supplemental Files.

## Supplemental Information

Supplemental information for this article can be found online at http://dx.doi.org/10.7717/peerj.10761#supplemental-information.

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
