# Peer review of "Rapid coral reef assessment using 3D modelling and acoustics: acoustic indices correlate to fish abundance, diversity and environmental indicators in West Papua, Indonesia"

_PeerJ, doi:10.7717/peerj.10761_

## Round 0.1 · original submission · Major Revisions

Dear authors,

I now received the comments of 2 referees about your paper, listed below. Both reviewers think that the paper could make a good contribution to PeerJ but both have a number of comments that need to be addressed. So I decided that I cannot accept your paper in the present state, but I encourage you to submit a revised version in which you take into account all the comments of the reviewers, paying particular attention at integrating the methods, as both reviewers highlight the lack of some important details. Moreover, an integration of the comparison with the available literature would help to put your work in the right context. Please carefully read and follow the PeerJ Resubmission Checklist for preparing your resubmission.

·

Basic reporting

no comment

Experimental design

no comment

Validity of the findings

no comment

Additional comments

The paper submitted by Peck et al. reports an interesting research regarding the monitoring of coral reefs through acoustic methods. Coral reefs have an exceptional ecological value that urgently needs to be preserved so that efficient monitoring tools are necessary for a proper assessment at local and regional scales. The research conducted by the authors tests the possibility to employ low-cost equipment and rapid analyses that could be informative to policymakers for conservation. This effort is highly valuable. However, I have several comments listed below. Among them, the most important one concerns the estimation of fish diversity through the unique estimation of total abundance. I am a bit surprised that diversity could be summarized by a single variable that does not take into account neither species diversity, nor phylogenetic diversity, nor functional diversity. Because fish abundance is the main result of the research the authors should justify the choice of this unique variable and they should describe as well properly how it was estimated on videos. This point should also be mentioned in the discussion.


Introduction

+ l50-51: The authors start their introduction with an economic justification of the importance of Indonesia's coral reef fisheries. I understand this is important to show that such ecosystems have a monetary value when you discuss with policy makers but here the text is addressed mainly to a scientific audience. I would therefore suggest to start rather with an ecological justification and mention later in the introduction or in the discussion the economic side of it.
+ l89-91: please add a reference for "The coral reef environment has a unique soundscape..."
+ l107: "health" might not be the most appropriate word. I would suggest "ecological state".
+ l108-109: I would suggest to keep the same order (visual then audio) than in the text above. I would then suggest to write: "do habitat metrics and acoustic indices from underwater audio and video correlate to coral cover and fish abundance?"

M&M
+ l115-117: I am not used to such sampling so I was not sure to understand the meaning of replicates here. Could the author clarify?
+ l131-133: The estimation of fish abundance does not seem to be fully explained. How was it estimated? Which method was used? Did someone tagged the videos? Who did it? Was it double-checked by two observers? What do the authors mean here by abundance? Is it the total number of individuals seen? If so, I am not sure that raw abundance is a good estimator of local diversity. Could the authors estimate species richness?
+ l132: "between 5 and 6 minutes after" should be replaced with "from minutes 5-6" as written in the next paragraph.
+ l139-140: The authors very interestingly used two approaches: beta indices (cumulative spectral dissimilarity) and alpha indices (ADI, AEI, etc). However the authors did not use both index families exactly in the same way. The alpha indices were computed by separating the 100-1200 and 1200-11000 Hz frequency bands separately, which is fine, but the beta index was computed on the full spectrum range. I would actually be very interested in seeing the results of the cumulative spectral dissimilarity index on each frequency band. The index might be more discriminant when applied to low frequency band where most of the acoustic activity occurs.
+ l144: I guess that the package tuneR (or seewave) should be mentioned here as Soundecology does not process sound extraction. Soundecology does not seem either to frequency filters so that the 100-1200 and 1200-11000 Hz bands could be separated with it. Actually, how were the two frequency bands separated? Which Fourier parameters were used to compute the indices?
+ l152: "sound pressure levels": RMS and roughness are not sound pressure level measures. RMS is an amplitude measurement (that could be related to sound pressure level if the recording system is calibrated) and roughness estimates the irregularity of a numeric series.
+ l153: Seewave -> seewave
+ l163-164: the author mention visual tools to check the validity of the models but the authors also should mention which assumptions these plots check for (normality and homoscedasticity of the residuals).
+ l167-168: It is a good idea to cite RStudio but RStudio is only a GUI for R. It is therefore more important to cite the version of R (R Core Team).

RESULTS
+ l171-72: this first sentence sounds more like a comment than a result. This could be transferred to the discussion.
+ l173-174: It was not clear in the M&M (l112-114) that there were 2 sites and that they will be compared. Actually the authors do not use exactly the same geographical names in the M&M and in the Results. Please clarify.
+ l173-186: The statistics used are not mentioned and were not introduced in the M&M section. I have to admit I do not understand the way the authors report the results: "F; 1, 29 = 21.02, p < 0.001". Is F the ANOVA statistic, and 1, 29 the degrees of freedom? Please use standard way to report statistics and mention the test used.
+ l191-198: the dB-RDA should be computed separately on the low and high frequency bands as for the alpha indices. This might then reveal significant results.

FIGURES
+ It would be nice to add an illustration showing some raw data: a picture fo the coral reef, an example of the 3D model, a spectrogram and a spectrum of a typical recording showing the low and high frequency bands. This could also be accompanied with a video and an audio sample.
+ Figure 1: why is "Manokwari" in bold face?
+ Figure 2: the legend does not fit with the figure. This is the plot of figure 4.
+ Figure 2-4: explain what "Ratio" means here.


DISCUSSION
+ The authors seem to use the same word "community" in a context of conservation policy and community ecology. Please specify each time this word is used.
+ The authors need to discuss about the reliability of the abundance estimator. Other studies using acoustics estimated species richness.
+ l288-293: this is correct comment, this is why I suggest to compute the index by frequency band.
+ l348: "pattern in AEI" -> "pattern in ACI".

TABLE
+ Table 1: "(AEI 0hz – 1200 Hz / AEI 0Hz – 11000Hz)" and others coud be replaced by "AEI low / AEI High". If not, please check Hz with a capital and space are properly written everywhere.
+ Table 2: how were defined the classes LOW and HIGH for fish abundance? This does not seem to be explained in the M&M section.

Jerome Sueur

Reviewer 2 ·

Basic reporting

a. Clear, unambiguous, professional English language has been used throughout the text.

b. Intro & background show context only partially. Some concepts are well explained and sufficient literature/ background information are provided (e.g. coral reefs status, management requirements, habitat metrics based on visual data). Other concepts are lacking in both literature and of depth of the explanation, in particular when it comes to acoustic indexes and vocal communities. Adding on these concepts will strengthen the paper and help the authors to better focus the significance and novelty of their findings.

Specific comments:

Line 67-85: Authors state “Two promising visual and audio approaches amenable to rapid and community-level data collection for coral reef monitoring including a) habitat metrics and b) use of acoustic indices as a proxy for vertebrate and invertebrate diversity”.
Although literature and context are well presented for the first method (a), the introduction lacks of context regarding acoustic indexes applied to marine environments (b). Only 2 literature references are provided here, one of which refers to terrestrial environments. A synoptic, more focused review of studies which have applied acoustic indexes to aquatic data, and of their findings, would surely strengthen the paper. For example, while some studies showed a correlation between taxonomic diversity and acoustic indexes, others showed that care should be taken when using acoustic indexes in different stages of their implementation, from calculation (dependence on settings, for example) to interpretation, at least for what concern marine data (e.g. which kind of acoustic information they detect? Abundances or diversity, or both? Are they sensitive to this ecologically relevant discrimination? etc). Adding a few lines in this sense would be beneficial. Also, it should be stated somewhere why indexes are getting so much attention, i.e. the allow the quick screening of large acoustic data without prior knowledge of the composition of the vocal community, on the contrary of manual analysis approach (which is extremely time consuming and requires skilled operators) or automated analysis based on sound type detection and recognition, e.g. Markov Chains (which requires previous knowledge on the target signals)

Suggested literature for indexes applied on aquatic data (mind that this list IS NOT exhaustive, but should provide a well-rounded picture)

 Davies, B. F., Attrill, M. J., Holmes, L., Rees, A., Witt, M. J., & Sheehan, E. V. (2020). Acoustic Complexity Index to assess benthic biodiversity of a partially protected area in the southwest of the UK. Ecological Indicators, 111, 106019.
 Elise, S., Urbina-Barreto, I., Pinel, R., Mahamadaly, V., Bureau, S., Penin, L., ... & Bruggemann, J. H. (2019). Assessing key ecosystem functions through soundscapes: A new perspective from coral reefs. Ecological Indicators, 107, 105623.
 Bolgan, M., Amorim, M. C. P., Fonseca, P. J., Di Iorio, L., & Parmentier, E. (2018). Acoustic Complexity of vocal fish communities: a field and controlled validation. Scientific reports, 8(1), 1-11.
 Bohnenstiehl, D., Lyon, R., Caretti, O., Ricci, S., & Eggleston, D. (2018). Investigating the utility of ecoacoustic metrics in marine soundscapes. J. Ecoacoustics, 2, R1156L.
 Elise, S., Bailly, A., Urbina-Barreto, I., Mou-Tham, G., Chiroleu, F., Vigliola, L., ... & Bruggemann, J. H. (2019). An optimised passive acoustic sampling scheme to discriminate among coral reefs’ ecological states. Ecological Indicators, 107, 105627.
 Bertucci, F., Parmentier, E., Lecellier, G., Hawkins, A. D., & Lecchini, D. (2016). Acoustic indices provide information on the status of coral reefs: an example from Moorea Island in the South Pacific. Scientific Reports, 6, 33326.
See also
 Desiderà, E., Guidetti, P., Panzalis, P., Navone, A., Valentini-Poirrier, C. A., Boissery, P., ... & Di Iorio, L. (2019). Acoustic fish communities: sound diversity of rocky habitats reflects fish species diversity. Marine Ecology Progress Series, 608, 183-197.

Line 93-94 The authors introduce the concept of acoustic communication in fish. No references are provided. Here, authors should provide references and expand their explanation.

Line 96-97 “biological community level acoustic indices as proxies for species richness and in tracking subtler habitat-dependent changes in community composition (Eldridge et al., 2018)”
Eldridge refers to avian communities, which are acoustically very different from aquatic communities. At line 102-103, on the other hand, there is no discussion about indexes limitations, which have been highlighted by other studies. I therefore refer the authors to my previous comment about acoustic indexes.

c. Literature is well referenced & relevant but its completeness could be improved (e.g. acoustic indexes, aquatic vocal communities)

d. Structure conforms to PeerJ standards

Experimental design

a. The article presents original primary research within Scope of the journal.

b. Research question are well defined, relevant & meaningful. However, the paper could be strengthened by adding a clearer explanation of how the research fills an identified knowledge gap.

c. Rigorous investigation was partially performed to a high technical & ethical standard.
Acoustic data were extracted from the GOPRO microphone, which is not as sensitive, nor omnidirectional, as a dedicated underwater hydrophone. This might have resulted in an underestimation of the vocal community dynamics and complexity. Furthermore, it lacks of temporal resolution, as it provides short-term data, depending on the GOPRO battery, which runs out quite quickly (on the contrary of dedicated acoustic loggers, which can provide months of continuous data). The authors should clearly state this limitation, showing they are aware of it, especially because in the discussion it seems that this point is, for them, a “novel” aspect of this study (line 250-253). There is a reason for which people use dedicated acoustic dataloggers for similar kind of studies; although I appreciate the (obvious) advantage of using the authors’ approach, a more nuanced and balanced discussion in this sense is required, especially if tools are suggested for management. The approach used by the authors has surely a very nice potential, but it would not provide information on long-temporal dynamics, just out of example.

d. Methods described with sufficient detail & information to replicate; partially.
The acoustic data lacks of important information, e.g. how many recordings were analysed? Why the authors chose to analyse only min 5-6 of a 10 min files? Which was the sample rate? And the Bit depth? In the analysis part… How was the mean frequency spectrum calculated, which FT size? Same applies to the calculation of indexes; which settings were used (e.g. temporal resolution? Frequency resolution?); RMS sound pressure levels were calibrated or not? Which was the unit of measurements (re dB or 1 dB re 1μPAa?) and so on.

Validity of the findings

a. The findings are novel but, because some methodological aspects are lacking and some results are not presented, it is difficult to ascertain at which extent the interpretation given by the authors is coherent. For example, at line 347-349 of the discussion, the authors state “the Acoustic Complexity Index ratio decreases with increase in fish abundance, also reflected in both frequency bands. Again, it may be reflecting anthrophony that correlates with higher fishing pressure or other anthropogenic impacts”. This is not clear because; a) the ACI “should” decrease in presence of anthropogenic noise, b) without knowing the settings used for its calculation, it is impossible to evaluate if the index was “tuned” at the best of its potential; c) there is mention to the vocal communities’ composition; e.g. how many sounds, how many sounds types, who were the species seen on site, do they belong to known vocal species? and so on

b. Significant results have been presented, but the lack of some methodological details makes interpretation difficult.

c. Conclusions are well stated, linked to original research question & limited to supporting results however they could benefit from wider comparison with other studies.

E.g. at line 388-390 the authors states “Acoustic indices offer insights to the status of biodiversity and function in reef environments, yet they remain rather blunt instruments”. They then suggest one alternative, but other alternatives have been suggested and should be mentioned/ considered (e.g. see Desidera’ et al. 2019 but there are others)

---

## Round 0.2 · Minor Revisions

As you can see, the reviews are positive, but reviewers suggested some modifications in the text and in two figures. I think that the specific issues raised by reviewer #3 can be easily addressed.

·

Basic reporting

The author took into account all the remarks of my first review including those requiring extra work.
I would just suggest to revise Figure 5 which looks as a simple screenshot of Audacity with all buttons around... Not very easy to read.

Experimental design

OK

Validity of the findings

OK

Additional comments

-

Reviewer 3 ·

Basic reporting

• Clear, unambiguous, professional English language used throughout the text.

• Intro & background show context in an exhaustive way, although the introduction would be greatly improved by providing extra background information when presenting acoustic indices (see below).

• Literature is well referenced but its completeness could be improved in the introduction section. Some works were incorrectly cited. Proper citations relating to coral reef soundscapes and tropical fish sound-producing species are needed (see below).

• Structure conforms to PeerJ standards.

• Figures are relevant and of high quality but may be improved (see below).

• R code and data sets available.

Specific comments
Lines 49-50: Please add one or more references for “Indonesia’s coral reefs support exceptional biodiversity, providing food security and other important ecosystem services to many millions of people.”
E.g. In the most recent report by FAO (2018) it is stated that in Indonesia fish contributed ≥50 % of total animal protein intake in 2015. Indonesia is also among the main exporters of fish and fish products
FAO. 2018. The State of World Fisheries and Aquaculture 2018 - Meeting the sustainable development goals. Rome. Licence: CC BY-NC-SA 3.0 IGO.
Line 78: “The principle method” -> The principal
Lines 93-96: Mis-used citations of the works by Bolgan et al. (2018 & 2020) and by Desiderà et al. (2019), which are not about coral reef soundscapes and tropical marine fishes. Please replace them and add more appropriate references for "The coral reef environment has a unique soundscape..., and avoiding predators".
Lines 97-98: I agree that sounds may be produced by soniferous fish species during all the activities stated here but the authors should either expand the number of references provided or cite a more comprehensive tropical-based acoustic work. Two citations (Picciulin et al 2019,2020) are about Mediterranean fish species thus mis-used considering the tropical context of the study.

Figures
Figure 1: I don’t know if the image I am reviewing has been updated for the second submission of the manuscript but there is still "Manokwari" in bold face. Moreover, in the map there are only 4 survey locations indicated while in the M&M section you refer to 8 sites. Are there 4 main study locations and 2 survey sites within each location? If so, you should specify it in the legend and in the text. What was the distance between the 2 sites within each location? You should add this detail in the text.

Experimental design

• The article presents original primary research within Scope of the journal.

• The research question is well defined, relevant & meaningful. It is stated how the research fills an identified knowledge gap.

• Rigorous investigation performed to a high technical & ethical standard.

• Methods described can be better detailed and explained so that the study can be replicated. At what time of the day were the surveys conducted? Were video and audio files recorded while SCUBA diving or free/skin diving? This would be crucial to know since SCUBA divers’ bubble noise would mask biological sounds. How many people were involved? In the discussion section (lines 296-298) you mentioned that “Each individual 10-minute replicate could be collected by 2 people in approximately 15 minutes, an important consideration if using SCUBA equipment that limits dive times” -> This detail (number of people involved) must be also mentioned in the M&M section. Were they always the same during data collection?

Specific comments
Lines 122-123: Which kind of management measures are in place in the protected reef systems? Is there any scientific evidence (report/study/paper) about the effectiveness of such protection measures and the greater ecological/conservation status of the protected reefs? Clarifying this may further support the statement you make later in the discussion section (Lines 400-403): “Soniferous fish are generally thought … in the healthier reef systems sampled.” (Assuming that these “healthier systems” are the protected reefs of Waisai and Mansuar).
Lines 124: Does South Manokwari coincide with Manokwari Seletan? If so, please be consistent with the name of survey sites/locations throughout the manuscript.
Line 125: You provided the overall number of collected replicates (34) but how many replicates did you collect within each of the 8 reef sites?
Line 133: “Corner of the transect” or corner of each of the 2m×2m quadrats? The term “transect” has not been defined earlier in the M&M section. Does it indicate the linear path at the same depth along which you deployed the replicates?
Line 144: How did you estimate the lengths of individual fish (>30 cm) from video frames?

Validity of the findings

• The findings are novel and significant.

• As far as I can see, all underlying data have been provided; they are robust, statistically sound, & controlled.

• Conclusions are well stated, linked to original research question & limited to supporting results.

Specific comments
Line 273: however -> with (?)
Line 296: quadrants -> quadrats
Line 304: specify that these are rapid short-term assessments -> “to support short-term reef assessments”
Lines 359-362: “It is worth noting… communities”. I have to admit that I find this sentence very long and I had to read it multiple times to get the point. Could you please clarify/rephrase?
Line 389: vocalizationS (?)
Line 402: increasing vocalizationS (?)
“healthier” -> more protected or pristine?
Lines 454-458: Honestly, I do not find this part very comprehensible, but it may be my fault. Would not be “develop” better than “development” in the sentence: “Data might be enhanced…. directly to components of reef function (Elise et al., 2019c).”? If so, please rephrase/clarify.

Additional comments

The research by Peck et al. is valuable as it investigates the use of low-cost technologies to provide rapid and key indications about the ecological status of coral reef ecosystems, whose study and conservation is of major importance nowadays. Moreover, I much appreciate the fact that the use of such technologies enables to archive material that can be re-analysed. I also consider very interesting the anecdotal evidence collected from local ecological knowledge about the use of snapping shrimps sounds as indicators of good fishing grounds (Lines 381-383).
Please find a few comments to improve the clarity of the paper.

---

## Round 0.3 · accepted · Accept

The revised version of the manuscript fulfills all the requests of the reviewers, and I think that it is ready for publication. Congratulation!